# Hypoxia and Cognitive Functions in Patients Suffering from Cardiac Diseases: A Narrative Review

**DOI:** 10.3390/jcm14196750

**Published:** 2025-09-24

**Authors:** Dominika Grzybowska-Ganszczyk, Zbigniew Nowak, Józef Alfons Opara, Agata Nowak-Lis

**Affiliations:** Faculty of Physiotherapy, Jerzy Kukuczka Academy of Physical Education, 40-065 Katowice, Poland; z.nowak@awf.katowice.pl (Z.N.); j.opara@awf.katowice.pl (J.A.O.); agata.nowak@awf.katowice.pl (A.N.-L.)

**Keywords:** ischemic heart disease, hypoxia, cognitive function, neuroprotection, myocardial infarction, Stroop test

## Abstract

**Background**: Cardiovascular diseases (CVD) are major contributors to global morbidity and mortality, and their association with cognitive impairment has gained increasing attention. Recent studies indicate that the prevalence of post-myocardial infarction (MI) cognitive impairment ranges from 22% to 37%, with attention being one of the most frequently affected domains. Moreover, novel approaches, such as normobaric hypoxic training in cardiac rehabilitation, show potential in improving both cardiovascular and cognitive outcomes. **Aim**: This narrative review aims to synthesize current evidence on the role of hypoxia in the development of cognitive dysfunction among patients with cardiac diseases, emphasizing shared mechanisms along the heart–brain axis. **Methods**: We performed a narrative search of PubMed, Scopus, and Web of Science databases using the keywords “hypoxia”, “cognitive impairment”, “myocardial infarction”, “heart failure”, and “CABG surgery”. We included original studies, reviews, and meta-analyses published between 2000 and up to the present in English. Priority was given to peer-reviewed human studies; animal models were included when providing mechanistic insights. Exclusion criteria included case reports, conference abstracts, and non-peer-reviewed sources. Narrative reviews, while useful for providing a broad synthesis, carry an inherent risk of selective bias. To minimize this limitation, independent screening of sources and discussions among multiple authors were conducted to ensure balanced inclusion of the most relevant and high-quality evidence. **Results**: Hypoxia contributes to cognitive decline through multiple pathophysiological pathways, including blood–brain barrier disruption, white matter degeneration, oxidative stress, and chronic neuroinflammation. The concept of “cardiogenic dementia”, although not yet formally classified, highlights cardiac-related contributions to cognitive impairment beyond classical vascular dementia. Clinical assessment tools such as the Stroop test, Trail Making Test (TMT), and Montreal Cognitive Assessment (MoCA) are useful in detecting subtle executive dysfunctions. Both pharmacological treatments (ACE inhibitors, ARBs) and innovative rehabilitation methods (including normobaric hypoxic training) may improve outcomes. **Conclusions**: Cognitive impairment in cardiac patients is common, clinically relevant, and often underdiagnosed. Routine cognitive screening after cardiac events and integration of cognitive rehabilitation into standard cardiology care are recommended. Future studies should incorporate cognitive endpoints into cardiovascular trials.

## 1. Introduction

Cardiovascular diseases (CVDs) are the leading cause of death and reduced quality of life worldwide. According to the World Health Organization (WHO), CVDs account for the primary cause of death globally. In 2019, an estimated 17.9 million people died from CVDs, representing 32% of all global deaths. Of these, approximately 85% were due to myocardial infarction and stroke. Among the 17 million premature deaths (under the age of 70) caused by non-communicable diseases in 2019, 38% were attributed to cardiovascular disease CVDs not only drive mortality but also contribute significantly to morbidity and disability-adjusted life years. Ischemic heart disease and heart failure represent the major contributors, yet increasing evidence demonstrates that patients with CVD are also at heightened risk of developing cognitive impairment (CI), ranging from mild cognitive dysfunction to dementia.

The heart and brain are functionally interconnected through the “heart–brain axis,” a complex, bidirectional system essential for sustaining life. Cardiac function and cerebral blood flow regulate brain metabolism and overall neurological performance. Therefore, cardiac dysfunction can lead to cerebral hypoperfusion, hypoxia, and subsequent cognitive decline, while neurological injury and neurodegeneration may adversely affect cardiac function through autonomic dysregulation. This conceptual framework provides a crucial background for understanding how myocardial infarction (MI), chronic heart failure (HF), and surgical interventions such as coronary artery bypass grafting (CABG) may trigger or accelerate cognitive decline [1,2,3].

Myocardial infarction is a life-threatening manifestation of coronary artery disease, characterized by sudden cardiac death. The risk of MI increases with age. In 2023, Salari et al. published a systematic review and meta-analysis on global MI prevalence. The analysis included 22 studies with a total sample size of 29,826,717 individuals under 60, revealing a global MI prevalence of 3.8%. In contrast, among individuals aged over 60 years (sample size: 5,071,185), the prevalence increased to 9.5% [4]. Sanchez-Nadales et al. examined clinical characteristics, management strategies, and outcomes in patients aged 80 years and older who presented with non-ST-elevation myocardial infarction (NSTEMI) in the United States between 2012 and 2018. Approximately 24.2% (151,472 of 625,916) of NSTEMI patients were aged 80 or older. In this older cohort, in-hospital mortality and cardiovascular complications were significantly higher compared to younger patients (odds ratio [OR] 1.79, 95% confidence interval [CI] 1.71–1.88, *p* < 0.001) [5].

Within this clinical context, hypoxia emerges as a central mechanism linking cardiac disease and cognitive decline. Recurrent episodes of cerebral and systemic hypoxia, whether due to reduced cardiac output, impaired autoregulation, or perioperative factors, are hypothesized to initiate a cascade of structural and functional brain changes. These include blood–brain barrier disruption, white matter injury, oxidative stress, and neuroinflammation—processes strongly implicated in the development of cognitive impairment and dementia.

This review aims to provide a comprehensive synthesis of current knowledge regarding hypoxia-related cognitive impairment in patients with cardiovascular diseases, integrating epidemiological data, pathophysiological mechanisms, clinical assessment tools, and therapeutic strategies. Narrative reviews, while useful for providing a broad synthesis, carry an inherent risk of selective bias (Ferrari, 2015) [6].

## 2. Cognitive Impairment Following Myocardial Infarction

Cognitive impairment (CI) is increasingly recognized as a frequent complication following myocardial infarction (MI). Sterling et al. (2019) reported that the prevalence of CI in patients with heart failure ranges from 20% to 80% [7]. This high prevalence led to the introduction of the term “cardiogenic dementia”, first mentioned in The Lancet in 1977, to describe cognitive decline resulting from cardiovascular conditions such as heart failure, myocardial infarction, and atrial fibrillation [8]. Coronary artery disease (CAD) has been associated with cognitive deterioration by reducing myocardial contractility, thereby decreasing cerebral blood flow and increasing the risk of dementia [9,10].

Myocardial infarction and the resulting systemic hemodynamic instability significantly contribute to the onset and progression of cardiogenic dementia. A multicenter study by Gharacholou et al. demonstrated that 29.8% of post-MI patients develop cognitive impairment (CI) [11]. A literature review by Burkauskas et al. identified several preliminary factors associated with cognitive function in patients with coronary artery disease, including coronary artery bypass grafting (CABG), the apolipoprotein E4 (APOE4) genotype, left ventricular ejection fraction (LVEF), pharmacotherapy, and various hormones and biomarkers. Importantly, novel CABG techniques have been shown to mitigate postoperative cognitive decline [12].

Recent prospective studies further expand our understanding of the natural course of post-MI CI. Kasprzak et al. (2023) reported that 37% of patients exhibited CI during hospitalization for acute MI, whereas this prevalence decreased significantly to 25% after six months (*p* < 0.001) [13]. This finding suggests that cognitive deficits may be transient or partially reversible, potentially due to neuroplasticity, optimized secondary prevention, or structured rehabilitation. Conversely, other longitudinal data indicate that CI may persist or even progress in patients with comorbidities such as diabetes, atrial fibrillation, or uncontrolled hypertension [14]. These discrepancies emphasize the heterogeneity of cognitive outcomes after MI and the necessity of long-term monitoring.

The relationship between reduced cardiac output and cognitive function has been extensively studied, yet findings remain inconclusive. For instance, Dikić et al. found no significant association between CI and LVEF in post-MI patients, thereby challenging the assumption that reduced LVEF directly leads to CI through cerebral hypoperfusion. Possible explanations include: (i) the contribution of non-hemodynamic mechanisms such as systemic inflammation or neurohormonal imbalance, (ii) the predominant influence of age and frailty as determinants of CI, and (iii) compensatory effects of cerebral autoregulation mitigating reduced cardiac output [15,16]. These findings highlight the multifactorial pathogenesis of CI, in which LVEF represents one—but not the sole—determinant.

Tirziu et al. further underscored the broad spectrum of neurocognitive dysfunction in heart failure, reporting that 30% to 80% of patients present with deficits in at least one domain, including memory, attention, learning, executive function, and psychomotor speed. Although coronary revascularization in ischemic HF may improve cardiovascular outcomes, it may also exacerbate neurocognitive impairment. Revascularization procedures such as CABG inherently carry the risk of delirium, postoperative CI, and stroke [17].

From the perspective of future research, improved control of confounding factors is warranted. Prospective cohort studies employing comprehensive neuropsychological assessment tools before, immediately after, and during long-term follow-up of myocardial infarction would provide more robust insights. Simultaneous monitoring of inflammatory markers, cerebral perfusion, and genetic factors (such as the APOE4 allele) could further elucidate the multifactorial mechanisms underlying CI in this patient population.

## 3. Cardiogenic vs. Vascular Dementia

The concept of “cardiogenic dementia” has gained increasing attention as a descriptive framework for CI arising from chronic cardiac dysfunction and hypoperfusion. While it shares pathophysiological overlap with vascular dementia, it highlights systemic cardiac dysfunction rather than focal cerebrovascular events.

Similarities.

Both involve impaired cerebral blood flow, white matter injury, and chronic hypoxia.

2.Differences.

Vascular dementia is typically driven by cumulative ischemic lesions (macro- or microinfarcts), whereas cardiogenic dementia results primarily from chronic cerebral hypoperfusion, neurohormonal dysregulation, and systemic inflammatory activation associated with HF or repeated ischemia.

Although “cardiogenic dementia” is not yet a formally recognized nosological category, it represents a clinically useful concept that may help identify high-risk patients and guide preventive strategies. Further research is warranted to clarify its diagnostic boundaries and relationship to established dementia subtypes [14,15,16,17,18,19,20,21] (Table 1).

## 4. Comparison: Cardiogenic Dementia vs. Vascular Dementia

Cardiogenic and vascular dementia share a common denominator–cerebral ischemia–but differ in the primary source of the problem:in cardiogenic dementia, the heart plays the central role (heart failure, arrhythmias, reduced cardiac output),in vascular dementia, the direct cause lies in cerebrovascular diseases and strokes.

In clinical practice, these two conditions often coexist and require comprehensive management of cardiovascular risk factors as well as causal treatment (management of heart disease, stroke prevention, and control of blood pressure, heart rhythm, and lipid levels).

## 5. Pathophysiology of Hypoxia and Cognitive Dysfunction

The brain is highly vulnerable to hypoxic injury due to its high oxygen demand and limited metabolic reserves. Chronic or intermittent hypoxia associated with cardiac diseases contributes to structural and functional alterations that underlie cognitive impairment. To enhance clarity, this section is organized into mechanistic subsections.

## 6. Blood–Brain Barrier (BBB) Disruption

Cardiogenic hypoxia increases the permeability of the BBB, facilitating the entry of pro-inflammatory cytokines and activation of microglia. This process contributes to neuroinflammation and neuronal injury. BBB dysfunction has been associated with the accumulation of pathological proteins such as β-amyloid, potentially accelerating the development of neurodegenerative disorders including vascular and Alzheimer’s dementia. Hypoxia also upregulates matrix metalloproteinases (MMPs), which degrade BBB components and further weaken its protective role [22]. Iadecola emphasized that chronic BBB disruption underlies the overlap between vascular dementia and cardiogenic dementia, highlighting its central role in hypoxia-mediated neurodegeneration [23].

## 7. White Matter Degeneration

Chronic cerebral hypoperfusion contributes to progressive white matter degeneration, particularly within the frontoparietal regions. Demyelination and axonal injury impair information-processing speed and executive functions. Population-based studies, such as SHIP-Trend-0, have demonstrated a significant correlation between hypoxic episodes and increased white matter hyperintensities (WMH) on MRI, suggesting that hypoxia accelerates brain aging and increases susceptibility to dementia [24]. Debette and Markus similarly reported that WMH are strongly associated with impaired executive function and reduced processing speed, reinforcing the link between chronic hypoxia and cognitive dysfunction [25].

## 8. Oxidative Stress and Mitochondrial Dysfunction

Hypoxia disrupts mitochondrial ATP production, leading to excessive generation of reactive oxygen species (ROS). Elevated ROS damage neuronal membranes, proteins, and DNA, thereby triggering apoptosis and neurodegeneration. Impaired mitochondrial activity also disrupts synaptic transmission and plasticity, negatively affecting memory and executive functions. Oxidative stress further amplifies neuroinflammatory processes, compounding hypoxia-related cognitive impairment [26]. Quaegebeur et al. provided additional evidence that mitochondrial dysfunction and oxidative stress represent key mechanisms driving neurodegeneration in chronic cardiac disease [27].

## 9. Neuroinflammation, a Double-Edged Sword

Myocardial infarction induces systemic hypoxia and inflammatory activation, both of which contribute to chronic neuroinflammation and accelerated neuronal injury. Activated microglia and astrocytes release cytokines and chemokines, which may initially serve protective functions but, when chronically elevated, promote synaptic loss and neuronal death [28]. This dual role has led to the description of neuroinflammation as a “double-edged sword”—simultaneously protective and deleterious depending on its duration and intensity. Comorbidities such as hypertension, type 2 diabetes, and dyslipidemia further intensify these processes, amplifying the risk of dementia.

## 10. Collateral Circulation and Angiogenesis

Interestingly, hypoxia also initiates compensatory mechanisms. The presence of well-developed collateral circulation in the territory supplied by an infarct-related artery significantly improves prognosis by limiting ischemia. Hypoxia stimulates angiogenesis and arteriogenesis, promoting the development of new vascular networks [29]. While this adaptation enhances oxygen delivery to ischemic regions, it may be insufficient to fully counteract hypoperfusion-related cognitive decline. Moreover, aberrant angiogenesis can create unstable vascular structures, paradoxically predisposing to microinfarctions. Thus, collateral vessel formation should be considered both a protective and potentially maladaptive process in the hypoxia–cognition axis.

## 11. Coronary Artery Bypass Grafting (CABG) and Cognitive Functions

CABG remains one of the most widely performed surgical interventions for advanced coronary artery disease. Despite its clear survival benefits and improvement of cardiac function, CABG is associated with a well-documented risk of neurocognitive complications. These complications are multifactorial, and their occurrence depends on surgical technique, patient vulnerability, and perioperative management.

## 12. Types of Cognitive Complications

Cognitive outcomes after CABG include two main entities:Postoperative Cognitive Dysfunction (POCD).

Subtle but measurable impairments in attention, memory, and executive functions, often detectable only by neuropsychological testing. POCD may persist for months after surgery and is associated with poorer long-term prognosis [30]

2.Delirium.

An acute, fluctuating disturbance of attention and awareness, typically arising in the first days after surgery. Delirium is especially frequent among elderly patients and those with pre-existing cognitive impairment, and it is linked with increased mortality and prolonged hospitalization [31].

Differentiating between POCD and delirium is crucial, given their different onset, duration, and underlying mechanisms.

## 13. Mechanisms of Neurocognitive Decline After CABG

Multiple overlapping mechanisms contribute to postoperative neurocognitive decline:Hypoxia and hypoperfusion. Pre-existing cerebral hypoxia from chronic cardiac disease increases neuronal vulnerability to intraoperative ischemia.Microembolic events. Aortic manipulation during cardiopulmonary bypass releases microemboli, often causing silent cerebral infarcts detectable on MRI.Systemic inflammation. Extracorporeal circulation induces a systemic inflammatory response that may cross the BBB, promoting neuroinflammation.Anesthetic neurotoxicity. Certain anesthetic agents may exacerbate neuronal apoptosis or interfere with synaptic plasticity, though evidence remains mixed.

Importantly, hypoxia should not be regarded as the sole driver of cognitive decline after CABG, but rather as a central factor interacting with embolic, inflammatory, and anesthetic-related mechanisms [32,33] (Table 2).

## 14. Role of the APOE4 Genotype

Genetic predisposition has been increasingly recognized as an important determinant of postoperative neurological outcomes. The apolipoprotein E ε4 (APOE4) allele, a well-established genetic risk factor for late-onset Alzheimer’s disease, has also been implicated in postoperative cognitive dysfunction (POCD) following cardiac and vascular surgery. Early evidence suggested that APOE4 carriers undergoing cardiopulmonary bypass surgery were at increased risk of cognitive decline. In a seminal report, Tardiff et al. observed that patients carrying the ε4 allele exhibited greater postoperative neurocognitive decline compared with non-carriers, despite no preoperative differences [34]. Similarly, Heyer et al. demonstrated that APOE4 carriers undergoing carotid endarterectomy had a significantly higher incidence of postoperative cognitive dysfunction at both early and 1-month follow-up compared with non-carriers [35]. These findings support the hypothesis that APOE4 confers increased vulnerability to perioperative cerebral hypoxia and inflammation. However, results have not been entirely consistent across surgical populations. In a prospective cohort of patients undergoing open aortic repair, Bryson et al. reported that the APOE4 genotype was not associated with either delirium or POCD after adjustment for confounding factors [36]. This variability likely reflects heterogeneity in surgical techniques, patient comorbidities, and outcome definitions. Mechanistically, APOE4 has been linked to impaired neuronal repair processes, reduced synaptic plasticity, and increased susceptibility to oxidative stress, providing a plausible biological basis for heightened risk of cognitive decline in surgical settings [37]. Taken together, these data suggest that preoperative identification of APOE4 carriers may enable risk stratification and inform the use of individualized neuroprotective strategies. Moreover, advances in surgical practice, such as off-pump coronary artery bypass grafting (CABG) and techniques to minimize cardiopulmonary bypass time, have been associated with lower rates of POCD in certain cohorts [38].

## 15. Surgical Techniques and Strategies to Reduce Risk

Recent refinements in surgical techniques and perioperative care aim to minimize the risk of cognitive complications.

Off-pump CABG (OPCAB). Avoids cardiopulmonary bypass, reducing embolic and inflammatory exposure.Minimal aortic manipulation techniques. Limit the risk of embolism and microinfarctions.Advanced cerebral perfusion monitoring (e.g., near-infrared spectroscopy). Enables real-time detection of hypoperfusion, allowing prompt corrective interventions.Neuroprotective protocols. Including optimized temperature control, anesthesia management, and perioperative pharmacological strategies (e.g., statins, antioxidants).

### 15.1. Cognitive Assessment Tools

Accurate detection of cognitive impairment in cardiac patients is crucial for prognosis and management. Cognitive decline after myocardial infarction (MI) or cardiac surgery may be subtle, and standardized neuropsychological tools are necessary to distinguish transient from persistent deficits. Neuropsychologists commonly employ a range of instruments, including the Mini-Mental State Examination (MMSE) [39], the Montreal Cognitive Assessment (MoCA) [40], the Clock Drawing Test (CDT) [41], the Benton Visual Retention Test (BVRT) [42], the Wechsler Adult Intelligence Scale (WAIS-R) [43], and comprehensive neuropsychological batteries.

### 15.2. The Stroop Test

The Stroop Color–Word Interference Test, first described in 1935 [44,45], remains one of the most widely used measures of executive function. Its relevance in cardiac patients arises from three critical features.

Sensitivity to frontal lobe dysfunction

Reduced perfusion in the prefrontal cortex, frequently observed in patients with cardiovascular disease, impairs Stroop performance [46]. These regions are central for executive functions such as planning, decision-making, and inhibitory control. Neuroimaging studies confirm that cardiac patients often demonstrate hypoperfusion in these areas, correlating with cognitive deficits.

2.Detection of subclinical cognitive impairment

The Stroop test detects early executive dysfunction that may not yet be visible on neuroimaging. Subtle impairments—such as slowed processing speed or reduced cognitive flexibility—can precede overt structural brain changes [47].

3.Prediction of cognitive decline

Longitudinal studies indicate that Stroop performance predicts future cognitive deterioration. For example, van Leijsen et al. demonstrated that patients with lower Stroop scores had a significantly increased risk of vascular dementia within five years after a cardiac event [48]. This aligns with findings that Stroop can act as an early biomarker of frontal lobe cognitive vulnerability.

### 15.3. Trail Making Test (TMT)

The Trail Making Test (TMT) complements Stroop by evaluating visuospatial attention, processing speed, and set-shifting ability. Part A assesses psychomotor speed, while Part B requires higher-order cognitive flexibility and executive control [49]. Impairments in TMT performance have been consistently reported in patients with chronic heart failure and post-CABG surgery, correlating with the burden of white matter hyperintensities [50].

### 15.4. Montreal Cognitive Assessment (MoCA)

The Montreal Cognitive Assessment (MoCA) is a brief yet sensitive screening tool that evaluates multiple domains, including memory, visuospatial ability, executive function, and attention. It has been validated in populations with mild cognitive impairment, heart failure, and cerebrovascular disease. Compared to MMSE, MoCA demonstrates superior sensitivity for subtle deficits particularly relevant to cardiac patients [40].

### 15.5. Integrating Cognitive Assessment

No single tool can comprehensively capture the complexity of cardiac-related cognitive impairment. A multimodal approach—incorporating Stroop, TMT, and MoCA—offers the most reliable assessment by targeting complementary cognitive domains. Standardized batteries are therefore essential not only in clinical care but also in research protocols assessing cognitive outcomes of cardiac interventions.

## 16. Treatment and Rehabilitation

The management of cognitive impairment (CI) in cardiac patients requires a multidimensional approach that integrates pharmacological and non-pharmacological strategies. The overarching goal is not only to optimize cardiovascular health but also to preserve or restore neurocognitive function.

## 17. Pharmacological Interventions

ACE Inhibitors and ARBs

Angiotensin-converting enzyme inhibitors (ACEIs) and angiotensin receptor blockers (ARBs) have shown beneficial effects on cognitive function by improving cerebral perfusion, reducing oxidative stress, and stabilizing the blood–brain barrier [51].

Observational studies reported lower incidence of cognitive decline in hypertensive and heart failure patients treated with ACE inhibitors [52].The PROGRESS trial demonstrated that perindopril combined with indapamide reduced the risk of cognitive decline and dementia in patients with cerebrovascular disease [53].

2.Other Cardiovascular Agents

Statins may reduce the risk of vascular dementia through lipid-lowering and anti-inflammatory effects [54,55].Beta-blockers show mixed effects: some studies suggest reduced cerebral perfusion and potential cognitive impairment, while others report protective effects against stress-related neurotoxicity.

## 18. Non-Pharmacological Interventions

Cardiac Rehabilitation (CR)

CR is a cornerstone of post-MI care and is recommended as early as possible, ideally during hospitalization [56].
Components include supervised physical exercise, education, risk factor management, nutrition counseling, pharmacotherapy, and psychological support.Benefits include improved cardiovascular performance, lower blood pressure, reduced systemic inflammation, enhanced left ventricular ejection fraction, and better endothelial function.Emerging evidence indicates that CR also stabilizes or improves cognitive performance [57].

2.Normobaric Hypoxic Training (NHT)

A novel adjunct to CR, NHT simulates altitude exposure during structured exercise.

Mechanisms. Stimulates angiogenesis, improves mitochondrial efficiency, and upregulates neurotrophic factors such as BDNF.Preliminary data suggest improved exercise tolerance and cognitive performance in ischemic heart disease.However, large randomized controlled trials are needed to confirm efficacy and ensure safety.

3.Cognitive training and psychosocial support

Structured cognitive rehabilitation—targeting attention, working memory, and executive functions—can enhance neuroplasticity and improve Stroop test performance [58].

In addition, addressing depression and anxiety through psychosocial support indirectly improves cognitive outcomes.

4.Optimization of risk factor management

Rigorous control of hypertension, diabetes, and dyslipidemia, alongside lifestyle modifications (diet, physical activity), reduces the risk of cognitive decline [55].

## 19. Integrative Approach

Optimal management of cardiac-related CI requires combining:Cardiovascular risk factor control.Evidence-based pharmacotherapy (e.g., ACEIs, ARBs, statins).Comprehensive CR (with potential integration of innovative strategies like NHT).Cognitive training and psychosocial interventions.

This holistic framework reflects the bidirectional heart–brain axis and aligns with the principles of precision medicine, aiming to preserve both cardiovascular and neurological health.

## 20. Conclusions and Future Directions

Cardiac diseases and hypoxia are closely linked to the development of cognitive impairment (CI), a condition that significantly affects prognosis, quality of life, and functional independence. This review synthesized current evidence on the heart–brain axis, underscoring hypoxia as a central mechanism that interacts with other pathophysiological factors, including oxidative stress, neuroinflammation, microemboli, and endothelial dysfunction. The concept of “cardiogenic dementia” has emerged to capture the systemic impact of cardiac pathology on cognition, although its nosological status remains under debate.

Epidemiological data indicate that nearly 30% of patients after myocardial infarction (MI) exhibit some degree of cognitive dysfunction, with advanced age being the strongest risk factor. While many studies demonstrate persistent cognitive decline, contradictory findings also exist. For example, Kasprzak et al. reported a reduction in CI prevalence six months after MI, suggesting that some deficits may be transient and partially reversible. Similarly, Dikić et al. found no significant correlation between left ventricular ejection fraction (LVEF) and cognitive status, highlighting the role of non-hemodynamic mechanisms such as systemic inflammation, genetic predisposition (e.g., APOE4), and microvascular pathology.

From a therapeutic perspective, both pharmacological (e.g., ACE inhibitors, ARBs, statins) and non-pharmacological strategies (e.g., structured cardiac rehabilitation, cognitive training, normobaric hypoxic training) show promise in mitigating CI in cardiac patients. Advances in surgical techniques, particularly in coronary artery bypass grafting (CABG), also aim to reduce postoperative cognitive dysfunction, though multifactorial mechanisms (hypoxia, microemboli, anesthetics, systemic inflammation) continue to present challenges.

## 21. Practical Clinical Implications

Routine cognitive screening

Standardized use of multimodal test batteries (Stroop, TMT, MoCA) in cardiology settings should be implemented to detect subtle deficits early.

2.Interdisciplinary care

Close collaboration between cardiologists, neurologists, neuropsychologists, and rehabilitation specialists is essential to address the bidirectional interactions between heart and brain health.

3.Tailored interventions

Risk stratification using genetic markers (e.g., APOE4), comorbidity profiles, and surgical risk factors should guide personalized therapy.

4.Surgical considerations

In CABG patients, both hypoxic and non-hypoxic mechanisms must be considered to minimize postoperative cognitive dysfunction.

## 22. Directions for Future Research

High-quality randomized controlled trials specifically designed with cognitive endpoints, testing both pharmacological and rehabilitation strategies.Longitudinal cohort studies to characterize the trajectory of cognitive decline after MI and CABG, distinguishing transient from persistent impairments.Biomarker-driven approaches (e.g., tau, β-amyloid, neurofilament light chain, APOE genotype) combined with advanced imaging modalities (functional MRI, PET) to identify patients at highest risk.Development of precision medicine frameworks that integrate cardiovascular, neurological, genetic, and sociodemographic data to design individualized interventions.Rigorous evaluation of innovative strategies such as normobaric hypoxic training, balancing its adaptive neuroprotective effects with the potential risks of exacerbating cerebral hypoxia.Translational studies using animal and cellular models of chronic hypoperfusion to inform mechanistic insights applicable to clinical practice.

## 23. General Conclusions

As understanding of the heart–brain axis evolves, it becomes clear that cardiovascular disease must be managed not only with the goal of prolonging survival but also with equal emphasis on preserving cognitive health. Addressing this dual burden requires an integrated clinical and research agenda, grounded in scientific rigor and patient-centered care, to develop targeted preventive strategies, enable early intervention, and ensure long-term quality of life for patients at risk of cardiogenic cognitive impairment.

## Figures and Tables

**Table 1 jcm-14-06750-t001:** Comparison of cardiac dementia and vascular dementia.

Feature	Cardiogenic Dementia	Vascular Dementia
Pathomechanism	Chronic hypoperfusion and cardioembolic events (atrial fibrillation, heart failure)	Stroke, small vessel disease, cerebral atherosclerosis
Onset	Slow, progressive	Often sudden (after stroke) or stepwise
Dominant symptoms	Memory and concentration impairment, general slowing	Executive dysfunction, impaired planning, spatial disorientation
Risk factors	Coronary artery disease, heart failure, arrhythmias	Hypertension, strokes, cerebral and carotid atherosclerosis
Neuroimaging	Nonspecific atrophy on MRI, hypoperfusion	Infarcts, vascular changes, leukoaraiosis
Course	Gradual, dependent on progression of heart disease	Stepwise, related to vascular events

**Table 2 jcm-14-06750-t002:** Cognitive complications after CABG – onset, symptoms, and clinical significance.

Type of Complication	Time of Onset	Main Symptoms	Clinical Significance
Postoperative delirium	1–7 days after surgery	Disorientation, disturbances of consciousness and attention, agitation or apathy	Prolonged hospitalization, higher risk of complications, increased perioperative mortality
Early postoperative cognitive dysfunction (POCD)	Weeks after surgery	Impaired short-term memory, attention, concentration, slowed thinking, executive deficits	Hinders cardiac rehabilitation, affects return to work
Long-term cognitive impairment	Months–years after CABG	Persistent memory and attention deficits, reduced learning ability, cognitive slowing	Reduced quality of life, risk of loss of independence
Dementia (accelerated or revealed by surgery)	Years after surgery (more common in elderly patients)	Progressive decline in memory, attention, executive functions; clinical picture of vascular or mixed dementia	Significant deterioration of social functioning and patient independence
Neuropsychiatric disorders (associated)	Early and late	Depression, sleep disturbances, reduced adaptability and information processing	Worsens prognosis, hinders participation in rehabilitation

## Data Availability

No new data were created or analyzed in this study.

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
