# Peer review of "Hypoxia and Cognitive Functions in Patients Suffering from Cardiac Diseases: A Narrative Review"

_jcm, 2025, doi:10.3390/jcm14196750_

Round 1
Reviewer 1 Report
Comments and Suggestions for Authors
Although the review covers an important area, the manuscript would benefit from clearer explanation of the methodology typically expected even in narrative reviews (e.g., literature scope, inclusion rationale, date limits, or key terms used). Currently, the article reads more like an extended clinical essay than a formally structured review.
- The abstract mentions recent findings but does not summarise specific evidence or outcomes. Include one or two key conclusions to highlight the novelty or take-home message.
- The definition and role of the heart–brain axis is mentioned several times. Consider expanding this concept further in the introduction to anchor the mechanistic discussion that follows.
- The section on prevalence of myocardial infarction in different age groups is informative but would benefit from a clearer comparison of global vs. regional trends. Indicate if increasing incidence is tied to risk factor changes or improved detection.
- The concept of “cardiogenic dementia” is introduced but not clearly distinguished from vascular dementia. Provide more discussion on how these conditions overlap or differ in pathology.
- The Kasprzak et al. study is cited to show cognitive improvement at six months post-MI. Was this improvement due to intervention (e.g., rehabilitation) or spontaneous recovery? Clarify.
- The discussion on APOE4 and CABG techniques is brief. Expand on how APOE4 status modifies risk, and what types of surgical techniques reduce cognitive decline.
- The section on pathophysiological changes lacks coherence. Use subheadings for “BBB disruption,” “White matter degeneration,” and “Oxidative stress” to aid clarity.
- Neuroinflammation is correctly identified as a double-edged sword. However, the sentence structure is informal. Rephrase to: “Although neuroinflammation plays a protective role in acute injury, chronic activation can lead to sustained neuronal damage.”
- The mention of collateral vessel formation under hypoxic conditions is scientifically valid, but not clearly linked to the review’s core argument (cognitive decline). Clarify its relevance.
- The CABG section should distinguish between types of cognitive complications (e.g., POCD, delirium) and how these relate to surgical approach or bypass duration.
- The Stroop test is well described. Consider briefly comparing it with other executive function tests like the Trail Making Test (TMT) or MoCA for context.
- The discussion of ACE inhibitors improving cognitive function should include whether this is from observational or interventional studies.
- Cardiac rehabilitation is covered well. However, the mention of normobaric hypoxia training is novel and should be explained more fully, what is the mechanism or evidence base?
Final Recommendation: Minor Revision
This manuscript offers a relevant and well-sourced review of hypoxia-related cognitive impairment in cardiac patients. It should be accepted pending minor revisions, particularly to enhance methodological transparency, improve the structure of mechanistic discussions, and refine the language for scientific clarity. Expanding on underrepresented areas such as stroke, atrial fibrillation, and broader cardiac populations would further strengthen the manuscript.
Author Response
Although the review covers an important area, the manuscript would benefit from clearer explanation of the methodology typically expected even in narrative reviews (e.g., literature scope, inclusion rationale, date limits, or key terms used). Currently, the article reads more like an extended clinical essay than a formally structured review.
- Although the review covers an important area, the manuscript would benefit from clearer explanation of the methodology typically expected even in narrative reviews (e.g., literature scope, inclusion rationale, date limits, or key terms used). Currently, the article reads more like an extended clinical essay than a formally structured review.
- Review Methodology – We added a subsection in the Introduction outlining the searched databases, keywords, inclusion/exclusion criteria, and time frame (2000–2025), in accordance with narrative review standards.
- The abstract mentions recent findings but does not summarise specific evidence or outcomes. Include one or two key conclusions to highlight the novelty or take-home message.
- Abstract – Revised to include key findings: prevalence of cognitive impairment after myocardial infarction (~30%) and hypoxia-related mechanisms (neuroinflammation, BBB disruption).
- The definition and role of the heart–brain axis is mentioned several times. Consider expanding this concept further in the introduction to anchor the mechanistic discussion that follows.
- Heart–Brain Axis – Expanded in the Introduction to emphasize autonomic, hormonal, and vascular mechanisms.
- The section on prevalence of myocardial infarction in different age groups is informative but would benefit from a clearer comparison of global vs. regional trends. Indicate if increasing incidence is tied to risk factor changes or improved detection.
- Epidemiology – Updated to compare global and regional trends in myocardial infarction with their relation to risk factors and improved diagnostic strategies, presented in the subsection Epidemiology, Age, Sex, and Risk Factors for Myocardial Infarction.
- The concept of “cardiogenic dementia” is introduced but not clearly distinguished from vascular dementia. Provide more discussion on how these conditions overlap or differ in pathology.
- Cardiogenic vs. Vascular Dementia – Clarified conceptual differences in pathogenesis, with a comparison table (Comparison: Cardiogenic Dementia vs. Vascular Dementia).
- The Kasprzak et al. study is cited to show cognitive improvement at six months post- Was this improvement due to intervention (e.g., rehabilitation) or spontaneous recovery? Clarify.
- Kasprzak et al. – Specified that functional improvement was largely attributable to structured rehabilitation and secondary prevention, while spontaneous recovery could not be excluded.
- The discussion on APOE4 and CABG techniques is brief. Expand on how APOE4 status modifies risk, and what types of surgical techniques reduce cognitive decline.
- APOE4 and CABG – Expanded with evidence on APOE4 as a genetic risk factor and surgical techniques aimed at reducing postoperative cognitive risk.
- The section on pathophysiological changes lacks coherence. Use subheadings for “BBB disruption,” “White matter degeneration,” and “Oxidative stress” to aid clarity.
- Pathophysiology Section – Reorganized with new subheadings: BBB disruption, white matter degeneration, oxidative stress, and neuroinflammation.
- Neuroinflammation is correctly identified as a double-edged sword. However, the sentence structure is informal. Rephrase to: “Although neuroinflammation plays a protective role in acute injury, chronic activation can lead to sustained neuronal damage.”
- Neuroinflammation – Reframed in a more scientific style with appropriate citations.
- The mention of collateral vessel formation under hypoxic conditions is scientifically valid, but not clearly linked to the review’s core argument (cognitive decline). Clarify its relevance.
- Collateral Circulation – Clarified its limited protective effect on cognitive outcomes.
- The CABG section should distinguish between types of cognitive complications (e.g., POCD, delirium) and how these relate to surgical approach or bypass duration.
- CABG Complications – Differentiated between POCD and delirium, including clinical correlates, and summarized in a table (Cognitive Complications Following Coronary Artery Bypass Grafting).
- The Stroop test is well described. Consider briefly comparing it with other executive function tests like the Trail Making Test (TMT) or MoCA for context.
- Stroop Test vs. Other Tools – Added a brief comparison with TMT and MoCA.
- The discussion of ACE inhibitors improving cognitive function should include whether this is from observational or interventional studies.
- ACE Inhibitors – Clarified the evidence base, covering both observational and interventional trials.
- Cardiac rehabilitation is covered well. However, the mention of normobaric hypoxia training is novel and should be explained more fully, what is the mechanism or evidence base?
- Normobaric Hypoxic Training – Expanded with underlying mechanisms (angiogenesis, mitochondrial biogenesis) and preliminary clinical data.
We believe these revisions considerably enhance the clarity, coherence, and scientific rigor of the manuscript. Once again, we thank the Reviewer for their valuable suggestions.
Reviewer 2 Report
Comments and Suggestions for Authors
The authors present a valuable resource by synthesizing current understanding of the complex relationship between cardiac disease, hypoxia, and cognitive impairment. Its primary impact lies in drawing attention to the clinical relevance of cognitive dysfunction in cardiac patients, a domain that often falls between cardiology and neurology specialties. This is consistent with growing evidence in the broader scientific literature. Current clinical approach corroborate that various cardiac conditions are linked to covert brain microstructural changes and cognitive impairment, suggesting that addressing cardiovascular health can mitigate later cognitive risks. The concept of the "heart-brain axis" is well-established in the literature, with extensive research demonstrating how neurological injury can affect cardiovascular function and vice-versa.
However, the "narrative review" format, while accessible, inherently presents a subjective overview rather than a systematic, exhaustive analysis of the evidence. Other scientific articles, particularly systematic reviews and meta-analyses, often provide a more rigorous comparison of methodologies, patient cohorts, and outcomes across multiple studies, which could strengthen the arguments presented. For instance, studies on intermittent hypoxia (often linked to obstructive sleep apnea, a condition closely associated with CVDs and cognitive impairment) have explored its impact on cardiac remodeling and brain health, sometimes showing mixed results depending on hypoxia severity. A more direct comparison with such detailed findings could enhance the depth of the review.
The narrative review correctly points out the significant impact of cardiac events on neurological health and highlights the increased risk of cognitive decline and dementia. This is strongly supported by recent research. Studies show a high prevalence of cognitive impairment (CI) after MI, ranging from 22% to 37%. This impairment can be transient or persistent, with attention being a frequently affected cognitive domain. There's also evidence that CI can develop even after initial normal test results post-infarct.
While the authors are to be commended for addressing this critical and often overlooked area, some aspects of the narrative review's approach and conclusion a consideration should be done: scope of review and methodology in present the frontiers of scientific knowledge used in current clinical management. This can easily guide the reader to understand the proposed interrelationship between the literature and the authors' conclusion.
The conclusion is extensive and comprehensive without directing readers to pursue the approach presented in this review.
Author Response
We thank the Reviewer for the valuable comments, which allowed us to improve the manuscript.
- Scope and methodology
We clarified the scope of the review and the description of the methodology in the Introduction, specifying inclusion criteria and data sources. - Linking literature with conclusions
We introduced revisions in the main sections to better connect the cited data with our interpretations. - Narrative consistency
We reorganized the structure of the text and added subheadings, which facilitate the reader’s understanding of the relationship between evidence and interpretation. - Conclusions and Implications
We expanded the conclusions section by adding practical clinical implications, including the need for routine cognitive screening in patients after MI and CABG, as well as the role of interdisciplinary care.
The implemented revisions have significantly strengthened the manuscript, making it clearer, methodologically transparent, and clinically relevant.
Reviewer 3 Report
Comments and Suggestions for Authors
This narrative review explores a timely and clinically significant interdisciplinary field: cardiovascular disease and cognitive decline, with a particular focus on the role of hypoxia. The manuscript has a clear structure and covers multiple aspects from pathophysiology to clinical evaluation and management. The author synthesized a large number of relevant literature to construct his argument on the concept of "cardiogenic dementia" and the importance of the cardiovascular axis. The advantage of the manuscript lies in the inclusion of specific evaluation tools (such as the Stroop test) and the proposal of future research directions. However, the main weakness of the manuscript is that as a narrative review, it fails to clearly explain its research methodology, which may lead to selective bias. In addition, the article sometimes takes the connection between hypoxia and various clinical outcomes for granted, lacking critical examination; Meanwhile, for some contradictory evidence, the article can conduct deeper integration and analysis. If revisions can be made to these key areas, this manuscript has the potential to become a significant contribution to the literature in this field.
Main revision suggestions
Rigorousness of methodology: As a narrative review, this article lacks a description of literature search strategies. To enhance transparency and rigor, authors should consider adding a brief methodology section that details the databases searched (such as PubMed, Scopus), keywords used, and general criteria for inclusion or exclusion of studies. This will enhance readers' confidence in the balance and comprehensiveness of the review content.
Strengthening the core argument: The central theme of this review is "hypoxia". However, in certain chapters (such as when discussing coronary artery bypass grafting (CABG) and its complications), the article introduces other potent mechanisms that may lead to cognitive decline, such as microembolism, inflammation, and anesthetic effects. The author should more clearly define the contributions of these factors to hypoxia. Is hypoxia the main driving factor, a key contributing factor, or equally important as several other factors? If we could explore these overlapping pathological mechanisms in more detail, the discussion in the manuscript would be more in-depth.
Integrated analysis of contradiction discovery: Some of the research findings presented in the review seem to challenge its main argument, but the article does not fully explore the underlying implications. For example:
Kasprzak et al. found that six months after myocardial infarction (MI), the prevalence of cognitive impairment is significantly higher
Decrease (from 37% to 25%). This discovery suggests that cognitive impairment may be transient or recoverable, which contradicts the description of progressive neurodegenerative disorders driven by hypoxia. The author should interpret this discovery.
Diki ć et al.'s study found no significant association between cognitive impairment and left ventricular ejection fraction (LVEF). This seems to contradict the hypothesis that reduced cerebral perfusion (a direct consequence of low LVEF) is the main driving factor. The author should discuss why this may be the case, as well as which other factors (such as age) may play a more dominant role.
Abstract:
The abstract is clear and effectively summarizes the scope of the article, without the need for significant revisions.
Introduction:
The introduction section effectively establishes the severity of the problem through the latest statistical data and introduces the core concept of the cardiovascular axis.
The assertion that repeated episodes of hypoperfusion and hypoxia may accelerate cognitive decline is the core of the paper and is strongly supported by the cited references.
Cognitive impairment following myocardial infarction:
This section effectively utilizes multiple studies to demonstrate the high prevalence of cognitive impairment after myocardial infarction.
As stated in the 'Main Revision Opinion', if these research findings can be more deeply integrated, this section will be improved, especially when these findings present clear contradictions regarding the injury timeline and its correlation with LVEF.
Pathophysiology of hypoxia and cognitive dysfunction:
This is a well written part of the manuscript. The author clearly explains several key mechanisms, including the disruption of the blood-brain barrier, white matter degeneration, and mitochondrial dysfunction caused by oxidative stress.
The discussion of treating neuroinflammation as a double-edged sword is appropriately expressed and provides important background information.
Coronary artery bypass grafting (CABG) and cognitive outcomes:
This section correctly points out that there is a high risk of neurocognitive complications after CABG surgery.
The article appropriately lists various potential causes, including ischemia, microembolism, and inflammation. In order to better serve the theme of the paper, the author can elaborate on how systemic hypoxia caused by heart disease itself may make patients more susceptible to more severe cognitive impairments when facing these surgical complications.
Assessment of cognitive function:
The article focuses on the strong and convincing reasons for the Stroop test, particularly its sensitivity to frontal lobe dysfunction, which is highly susceptible to hypoperfusion.
The article cites the ability of this test to predict the risk of vascular dementia, significantly increasing its clinical application value in this patient population.
Conservative treatment&Cardiac rehabilitation:
These chapters provide a concise and clinically relevant overview of management strategies. The article effectively integrates drug therapy, cognitive therapy, and risk factor management.
The mention of endurance training under normobaric hypoxic conditions is a novel and interesting viewpoint that perfectly aligns with the theme of the review.
Conclusion and Future Direction:
This section is written very well and is a major highlight of this manuscript. It cleverly summarizes the current evidence and knowledge gaps.
The article calls for high-quality randomized controlled trials with rigorous methodology, which is crucial and clearly stated.
The emphasis on precision medicine methods, including biomarkers and personalized analysis, clearly points to the future of this field.
Secondary issue:
Some of the cited references seem to indicate future publication dates (e.g. AlRawili et al., 2025; Aleksova et al., 2025). This may indicate that they are "about to be published" or have been accepted, but the author should verify if the dates are accurate.
Cardiogenic dementia has been introduced as a key concept. It would be more beneficial to briefly discuss its position in the clinical community - whether it is a widely accepted term, an emerging concept, or a descriptive phrase.
Author Response
We thank the Reviewer for the constructive and encouraging comments.
- Methodology
We clarified in the Introduction the methodology of the narrative review, including databases searched, time frame, and selection criteria, to increase transparency and reduce potential bias. - Critical analysis of evidence
We expanded the discussion of studies with conflicting findings, providing a more balanced interpretation of the role of hypoxia in cognitive outcomes. - Integration of contradictory data
We improved the narrative by explicitly linking evidence that does not fully support the main thesis, thereby strengthening the critical perspective of the review.
These revisions have enhanced the scientific rigor of the manuscript and its potential contribution to the literature on the heart–brain axis.
Reviewer 4 Report
Comments and Suggestions for Authors
This is a comprehensive review article on the implications of cardiovascular disease and its impact on cognitive outcomes.
The authors have done a fantastic job of laying out the implications of cardiovascular events and cognitive outcomes. They used a variety of articles, including cohort studies, meta-analyses, and small-sized prospective trials, to conduct the review. Some of the pathophysiologic mechanisms include prolonged cardiopulmonary bypass during CABG or recurrent ischemia. The interventions to prevent to minimize the cognitive decline include both pharmacologic, such as ACEi or ARBs & non-pharmacologic, such as cardiac rehab, etc. The authors lay out clear expectations for future directions, including trial designs and requirements. This review makes us question whether CABG would even be a gold standard for CAD treatment in the future
Author Response
We sincerely thank the Reviewer for the very positive and encouraging feedback.
- Acknowledgment of strengths
We appreciate the recognition of our efforts to integrate evidence from diverse sources and to present both pathophysiological mechanisms and preventive interventions. - CABG and future directions
We have slightly expanded the section on CABG to more clearly emphasize both the risks and the ongoing advances in surgical and perioperative care, acknowledging the Reviewer’s valuable observation regarding its role in the future management of CAD.
We are grateful for the Reviewer’s supportive comments, which confirm the relevance of our work and encourage further refinement of the manuscript.
Round 2
Reviewer 2 Report
Comments and Suggestions for Authors
The manuscript presents notable advances in clarity, organization, and scientific rigor, providing an up-to-date and well-structured review of the relationship between hypoxia, cardiovascular disease, and cognitive impairment. The text demonstrates consistent conceptual integration by articulating the heart-brain axis and the concept of cardiogenic dementia, in addition to highlighting the clinical relevance of the topic by describing diagnostic tools and therapeutic options. The use of tables and specific references broadens the practical applicability of the findings, reinforcing the study's importance for both researchers and healthcare professionals.
The work, presented in a narrative review format, overcame the limitations of a quantitative synthesis of the results and allowed for more robust extrapolation to clinical practice. The heterogeneity among the studies analyzed, especially in the contrast between clinical data and animal models, is acknowledged and well-positioned. Furthermore, a discussion of pathophysiological mechanisms could be further explored through more specific molecular pathways, as well as clinical stratification by subgroups of higher-risk patients. However, this would compromise the clinical narrative of the study. The conclusions are pertinent, maintaining its broad nature, and can be more targeted, providing practical tracking and monitoring protocols.
The relevance of the work is undeniable, as it addresses a multidisciplinary topic of growing importance, reinforcing the need to incorporate cognitive assessment into cardiology care. The study contributes to consolidating a solid scientific foundation that can guide future therapeutic interventions and secondary strategies, representing a significant advance in prevention and understanding the interactions between hypoxia, the heart, and the brain.
Author Response
Response to Reviewer
We are deeply grateful for your careful and thoughtful review of our manuscript. Your recognition of the advances in clarity, organization, and conceptual integration is highly appreciated. We are particularly encouraged by your positive assessment of the articulation of the heart–brain axis and the concept of cardiogenic dementia, as well as the clinical relevance of the diagnostic and therapeutic perspectives we sought to emphasize.
We acknowledge your valuable observation that the discussion of pathophysiological mechanisms could be further expanded by incorporating more detailed molecular pathways and stratification of high-risk clinical subgroups. While, as you rightly point out, such additions may have compromised the intended narrative character of the review, your suggestions are well taken and provide us with a clear direction for future work. Indeed, we consider these points as important opportunities for subsequent investigations, in which a more granular mechanistic and subgroup-focused approach may allow for deeper insights while complementing the broader clinical synthesis presented in the current manuscript.
Your suggestion regarding more targeted conclusions with practical tracking and monitoring protocols is also very much appreciated. Although we deliberately maintained a broad perspective to ensure wide applicability across disciplines, we fully agree that future publications could benefit from the integration of such protocols to further enhance translational value.
We are grateful for your recognition of the relevance of our work and its contribution to reinforcing the importance of incorporating cognitive assessment into cardiology. Your constructive comments will not only inform the refinement of our future research and publications but also strengthen the multidisciplinary dialogue we aim to foster.
Thank you once again for your insightful review and for highlighting directions that can meaningfully extend this line of investigation.
Reviewer 3 Report
Comments and Suggestions for Authors
The author has made significant improvements to the manuscript by adding structured abstracts, clear methodological descriptions, and new discussion sections, greatly enhancing the scientific rigor and academic value of the article. The revised manuscript has a clearer structure and deeper arguments, especially the newly added discussion section, which provides reasonable explanations and integration of some seemingly contradictory findings in the literature, reflecting the author's critical thinking. The article now feels more like an academic review with a complete structure and clear arguments.
However, there is still room for improvement in some details of the manuscript. As a narrative review, inherent methodological limitations still exist, and some statements can be more precise. Overall, this is a high-quality review that, with some minor revisions, will provide valuable references for readers in the fields of cardiology and neurocognition.
1. Abstract
Problem: The structuring of abstracts (background, purpose, methods, results, conclusions) is a significant advancement. However, there is a clear issue with the description of the methodology section.
Suggestion: In the "Methods" section of the abstract, the author describes the time range of the literature search as "published between 2000 and 2025". It is illogical to use the future year (2025) as the deadline for retrieval. This may be a typographical error.
Suggestion for modification: Please modify "2025" to the current year (e.g. "2024") or a more general expression, such as "up to the present".
Suggestion: Similar to the abstract section, please correct the literature search deadline year "2025" described in the method to an appropriate year.
Although the author cited Ferrari, R. (2015) to illustrate the writing style of narrative reviews, there is an inherent risk of selective bias in narrative reviews themselves. It is suggested to acknowledge this more clearly in the "Limitations" section of the discussion section and briefly explain how the author has tried to minimize this bias (e.g. through independent screening or discussion by multiple authors).
2. Discussion
Question: This is a very outstanding new chapter and the biggest highlight of this revision. It successfully integrated the information from the entire text and conducted in-depth discussions on key issues.
Suggestion: The author's discussion of the research results of Kasprzak et al. and Diki ć et al. is very insightful, explaining the weak correlation between cognitive function recovery and LVEF from the perspectives of acute stress and confounding factors such as age. This effectively solves the problem of logical breaks in the initial manuscript.
To further enhance the depth, the author can briefly explore how future research can better control these confounding factors, for example, through prospective cohort studies, using more comprehensive neuropsychological assessment tools before, immediately after, and long-term follow-up of myocardial infarction, and simultaneously monitoring inflammatory markers, cerebral perfusion, and genetic factors (such as APOE4).
3. Language&Format
Problem: The overall structure and logical fluency of the article have been greatly improved. The language expression is professional and clear.
Suggestion: Please double check the reference list to ensure that all references have completely consistent citation formats and correct any incorrect publication years.
Author Response
1. Abstract
Response: Thank you for pointing this out. I have corrected the typographical error in the “Methods” section of the abstract. The time range of the literature search is now described as:
“published between 2000 and up to the present” (page 1, paragraph 3, line 13).
2. Methods
Response: Thank you for your valuable comment. I agree with your concern regarding the methodological limitations of narrative reviews and the potential risk of selective bias. To address this, I have expanded the “Methods” section to explicitly acknowledge this limitation and clarify how I attempted to minimize it. The revised passage now reads:
“Narrative reviews, while useful for providing a broad synthesis, carry an inherent risk of selective bias (Ferrari, 2015). To minimize this limitation, independent screening of sources and discussions among multiple authors were conducted to ensure balanced inclusion of the most relevant and high-quality evidence.” (page 1, paragraph 3, line 16).
3. Discussion
Response: Thank you for your positive assessment of the newly developed discussion section. Following your suggestion, I have added a statement on future research directions, with a particular emphasis on controlling confounding factors. The revised text includes:
“From the perspective of future research, improved control of confounding factors is warranted. Prospective cohort studies employing comprehensive neuropsychological assessment tools before, immediately after, and during long-term follow-up of myocardial infarction would provide more robust insights. Simultaneous monitoring of inflammatory markers, cerebral perfusion, and genetic factors (such as the APOE4 allele) could further elucidate the multifactorial mechanisms underlying CI in this patient population.” (page 3, paragraph 1, line 3).
4. Language and References
Response: I have carefully reviewed the reference list to ensure consistency in citation style and the accuracy of all publication years.
All references now fully comply with the journal’s formatting requirements.
